# Signal Detection by Sensors and Determination of Friction Coefficient During Brake Lining Movement

**DOI:** 10.3390/s24248078

**Published:** 2024-12-18

**Authors:** Leopold Hrabovský, Vieroslav Molnár, Gabriel Fedorko, Nikoleta Mikusova, Jan Blata, Jiří Fries, Tomasz Jachowicz

**Affiliations:** 1Department of Machine and Industrial Design, Faculty of Mechanical Engineering, VSB-Technical University of Ostrava, 17. listopadu 2172/15, 708 00 Ostrava-Poruba, Czech Republic; leopold.hrabovsky@vsb.cz (L.H.); jan.blata@vsb.cz (J.B.); jiri.fries@vsb.cz (J.F.); 2Faculty of Manufacturing Technologies, Technical University of Kosice with a Seat in Presov, Bayerova 1, 080 01 Presov, Slovakia; vieroslav.molnar@tuke.sk; 3Faculty of BERG, Technical University of Kosice, Park Komenskeho 14, 040 01 Kosice, Slovakia; nikoleta.mikusova@tuke.sk; 4Faculty of Mechanical Engineering, Lublin University of Technology, 36 Nadbystrzycka St., 20-618 Lublin, Poland; t.jachowicz@pollub.pl

**Keywords:** force sensor, torque sensor, friction coefficient, laboratory device, shoe brake

## Abstract

This article presents a laboratory device by which the course of two signals can be detected using two types of sensors—strain gauges and the DEWESoft DS-NET measuring apparatus. The values of the coefficient of friction of the brake lining when moving against the rotating shell of the brake drum were determined from the physical quantities sensed by tensometric sensors and transformed into electrical quantities. The friction coefficient of the brake lining on the circumference of the rotating brake disc shell can be calculated from the known values measured by the sensors, the design dimensions of the brake, and the revolutions of the rotating parts system. The values of the friction coefficient were measured during brake lining movement. A woven asbestos-free material, Beral 1126, which contained brass fibers and resin additives, showed slightly higher values when rotating at previously tested speeds compared to the friction coefficient values obtained when the brake drum rotation was uniformly delayed. The methodology for determining the friction coefficient of the brake lining allowed the laboratory device to verify its magnitude for different friction materials under various operating conditions.

## 1. Introduction

Friction is an “ordinary” physical/mechanical phenomenon encountered in everyday life. It can be defined as a mutual contact-based interaction between different bodies that prevents their relative motion. Liang XM et al. [1] conducted experiments that revealed that the linear relationship between friction force and average load always holds. However, for the relation between friction force and actual contact area, the linearity holds only at the loading stage, while it fails at the unloading stage. Holmberg and Erdemir [2] report that wear is more critical than friction because it can lead to catastrophic and operational failures that adversely affect productivity and, thus, costs. Hsu S. et al. [3] state that controlling friction becomes essential in seeking sustainable technologies. Friction, after all, is an indicator of energy efficiency. If it is possible to reduce the unnecessary parasitic energy losses and increase the current energy efficiency, it will give us time to develop alternative energy sources. Krc et al. [4] presented a database of shear friction test results collected from the literature and analyzed using the approaches in the *PCI Design Handbook*. Santos and Júlio [5] presented a literature review on design expressions for shear friction, which they described chronologically, starting with proposals from the earliest research studies, precursors of the theory, until the most recent studies incorporated in the newest *fib* Model Code. Blau [6] deals with the measurement and use of static and kinetic coefficients of friction, discusses their usefulness, and describes sources of frictional resistance in terms of shear localization. Blau and Jolly [7] investigated whether the wear of brake lining materials can be measured in shorter-term laboratory tests and, if so, determined to what extent the relative ranking of several lining materials’ wear resistance depends on the testing method. Ertan and Yavuz [8] experimentally investigated the brake lining composition for the effects of manufacturing parameters on tribological properties and to obtain optimal manufacturing parameters for improved tribological behavior. Laguma-Camacho et al.’s study [9] described the wear mechanisms involved in disc and shoe pads (bodies). Using theoretical mechanics, Dosaev [10] solved the classical problem of the contact between a shoe and a wheel. They also analyzed each element’s elastic and viscous properties, simulating the material’s compliance in the contact area between the wheel and the brake shoe. They determined that elastic force acts on the shoe from the side of the pusher, pressing the shoe against the wheel. Teoh et al. [11] show the development of a minimal model of a drum brake squeal under binary flutter instability, which is caused by the velocity-independent friction coefficient.

The function of the brake lining (or the entire brake system) is to reduce how long it takes for a moving transport device or means of transport to stop by exerting friction between the brake lining and the rotating brake drum. In the work of Aleksendrić and Barton [12], the synergistic effects of the friction material’s properties, defined by its composition and manufacturing conditions, and the brake’s operating regimes on the disc brake factor *C* variation were modeled using artificial neural networks. Yang and Chen [13] created an inverse algorithm based on the conjugate gradient method and the discrepancy principle and used it to estimate the unknown space- and time-dependent heat flux of the disc in a disc brake system. Their calculations were based on temperature measurements taken within the disc. 

In several engineering applications, friction is an undesirable phenomenon, and efforts are made to eliminate it; in the case of brake linings, friction is an essential prerequisite for the safe operation of transport equipment.

The methodology for measuring the coefficient of shear friction of the brake lining at the moment of brake disc start-up is discussed in [14,15]. The value of the shear friction coefficient depends mainly on the surface condition and the materials’ roughness [16]. For cases where Amonton friction laws apply [17,18], the values of the factor range from zero (practically no friction) to several times higher than one, indicating that the frictional force is many times greater than the perpendicular pressure. Since 2012, it has been known and experimentally proven that in a specific small range of perpendicular pressures, the coefficient of shear friction can be negative (Gao et al. [19]).

A distinction is made between shear friction and friction at rest (static friction). Hrabovský and Janek [20] presented an experimental measurement of the mining equipment’s motion resistance. Friction during movement (the magnitude of the friction force in kinematic friction) does not depend on speed. The experiments realized by Lee et al. [21] classified different forms of friction.

It is generally known that the magnitude of the coefficient of friction on the standard contact surface of two bodies cannot be measured directly. Burris and Sawyer [21] reviewed the challenges of low-friction measurements and presented a robust reversal technique that eliminates misalignment bias. Schmitz et al. [22] investigate the experimental uncertainty associated with friction measurements obtained by following the guidelines prescribed in international standards.

The friction coefficient is calculated from two measured forces—compressive normal force and tensile force. The friction coefficient at the slip point, or during the slippage of the conveyor belt on the rubber or steel casing of the driving drum, was determined for two states of the surfaces in contact. Experimental measurements on a laboratory machine determined four values of friction coefficients for two types of drum surfaces and for two states of contact surfaces, which were compared with the recommended standard values. The measured values reached higher than those given by the CSN standards (Hrabovský et al. [23]). Straffelini [24] defines the transmission ratio of the acting forces as the coefficient of static friction. In general, it is independent of the nominal contact area, while it can rely on the properties of the opposing materials and their surfaces.

A closer investigation of the mechanics of the body’s contact, including numerical modeling, experiments, and determination of the actual contact spatial area, including the macro/micro/nano view and the friction coefficient and friction models, is given by [1,2,3,25,26,27,28,29,30,31,32,33].

The paper presents a laboratory device for measurements (using two sensors—strain gauges and DEWESoft DS-NET measuring apparatus). Based on the brake design dimensions and the speed of the rotating system, the friction coefficient of the brake lining on the circumference of the brake disc rotating casing is determined and calculated. The methodology that was created enables the laboratory device to verify the magnitude of the friction coefficient for different friction materials under various operating conditions.

## 2. Materials and Methods

The laboratory device (see Figure 1) is formed by a welded steel construction 1 of closed profiles with a cross-sectional dimension of 30 × 40 mm. A foot electric motor 2 [34] (type AP 112M-S3, power P_e_ = 1.5 kW, revolutions n_e_ = 710 min^−1^) is attached to the structure 1 using screw connections. A torque sensor 6 [35] is installed between the shaft 8 and the electric motor 2. The shaft 8 is housed in three plummer blocks 9 (type SN 507 [36]), which are attached to the Jäckl by screw connections 10. The brake disc 4 (ϕ200 mm, width 85 mm) of the disc brake is placed on the shaft 8 using a tight spring (8e7 × 7 × 50 [37]). A flywheel 3 (ϕ320/36 ϕmm, width 28 mm) is mounted on the threaded part of the shaft 8.

Figure 2 presents a 2D assembly sketch of the laboratory device’s structural design, adapted from a volumetric 3D model created in the SolidWorks 2012 ×64 Edition SP05 software environment [39] and made in the AutoCAD 2010 software environment [40].

The standard plummer block in a laboratory device is used to fasten the bearings to the bearing structure. The plummer block 9 (see Figure 1 and Figure 2) is a split version of the standing plummer block in the laboratory device used to place the shaft 8 (see Figure 3a; its upper part is removable, significantly simplifying assembly and maintenance).

Equation (1) gives the moment equation of forces acting on point A (see Figure 4b) on the brake lever 2, provided that G [N] is the weight 7 acting on the arm e [m] of the brake lever, α[deg] is the angle of inclination of the threaded rod 6 of the brake, F_M_ [N] is the force in the threaded rod 6 of the brake derived by the weight G [N] through the brake lever 2 of length e [m], or the tensile force acting in the threaded rod 6 (see Figure 4c), detected by the force sensor 5.
(1)$$\sum M_{A}=0\;N\cdot m\;\Rightarrow {\;F}_{M}\cdot \cos(\alpha )\cdot d\;-\;G\cdot e=0\;\Rightarrow {\;F}_{M}=\frac{G\cdot e}{d\cdot \cos(\alpha )}\;[N]$$

By using the moment equation of the forces acting on point B (see Figure 4b), it is possible to express the magnitude of the pressure force N_1_ [N] (2) of the left brake shoe 3 (see Figure 4a) against the brake disc 4.
(2)$$\sum M_{B}=0\;N\cdot m\;\Rightarrow {\;N}_{1}\cdot a{\;-\;F}_{M}\cdot \cos(\alpha )\cdot l=0\;\Rightarrow {\;N}_{1}=\frac{F_{M}\cdot \cos(\alpha )\cdot l}{a}\;[N]$$

If (1) is added to Equation (2) after F_M_ [N], the magnitude of the pressure force N_1_ [N] of the left brake shoe 3 against the brake disc 4 can be expressed by Equation (3).
(3)$$N_{1}=\frac{F_{M}\cdot \cos(\alpha )\cdot l}{a}=\frac{G\cdot e\cdot \cos(\alpha )\cdot l}{d\cdot \cos(\alpha )\cdot a}=G\cdot \frac{e\cdot l}{a\cdot d}\;[N]$$

From the moment equation of the forces acting on point C (see Figure 4b), it is possible to express the magnitude of the pressure force N_2_ [N] (4) of the right brake shoe 3 (see Figure 4a) against the brake disk 4.
(4)$$\sum M_{C}=0\;N\cdot m\;\Rightarrow {\;F}_{M}\cdot \cos(\alpha )\cdot l{\;-\;N}_{2}\cdot a=0\;\Rightarrow {\;N}_{2}=\frac{F_{M}\cdot \cos(\alpha )\cdot l}{a}\;[N]$$

If (1) is added to Equation (4) after F_M_ [N], the magnitude of the pressure force N_2_ [N] of the right brake shoe 3 against the brake disk 4 can be expressed by Equation (5).
(5)$$N_{2}=\frac{F_{M}\cdot \cos(\alpha )\cdot l}{a}=\frac{G\cdot e\cdot \cos(\alpha )\cdot l}{a\cdot d\cdot \cos(\alpha )}=G\cdot \frac{e\cdot l}{a\cdot d}\;[N]$$

In double-jaw brakes, two jaws are placed against each other on pivots. The braked circumferential force F_u_ [N] and the braked moment M_u_ [N·m] can be expressed, assuming the same magnitude of both pressure forces of the brake jaws N_1_ [N] (3) and N_2_ [N] (5), according to Equation (6).
(6)$$F_{u}=\left(N_{1}{+N}_{2}\right)\cdot f=2\cdot F_{M}\cdot \cos(\alpha )\cdot f\cdot \frac{l}{a}=2\cdot G\cdot f\cdot \frac{e\cdot l}{a\cdot d}{\;[N];\;M}_{u}{=F}_{u}\cdot \frac{D}{2}{=F}_{M}\cdot \cos(\alpha )\cdot f\cdot \frac{l\cdot D}{a}=G\cdot f\cdot \frac{e\cdot l\cdot D}{a\cdot d}\;[N\cdot m]$$

The implemented measurements were performed on a test device; see Figure 3. The electric motor 2 is used to start up the flywheel 3 to the desired angular speed ω [rad·s^−1^]; once this speed is reached, the motor is disconnected, and the double-jaw brake is activated. Flywheel 3 consists of thin (thickness of 28 mm) discs of constant dimensions. The addition of discs allows changing the weight of the flywheel and thus the moment of inertia of the moving (rotating) weights of the test device J [kg·m^2^] (7). The programming device allows setting the desired operating mode, i.e., angular acceleration ε [rad·s^−2^] (7).
(7)$${J=J}_{M}+J_{s}+J_{B}\;[\mathrm{kg}\cdot m^{2}];\;\varepsilon =\frac{d\omega }{\mathrm{dt}}=\frac{\omega }{t_{b}}=\frac{2\cdot \pi \cdot n_{M}}{t_{b}}\;[\mathrm{rad}\cdot s^{-2}]$$
where J_M_ [kg·m^2^] is the moment of inertia of the engine (J_M_ = 0.0122 kg·m^2^ [34]), J_S_ [kg·m^2^] is the moment of inertia of the flywheel (8), and J_B_ [kg·m^2^] is the moment of inertia of the brake disc (J_B_ = 0.06 kg·m^2^, read from the SolidWorks 3D modeler [39]),
(8)$$J_{s}=\frac{1}{2}\cdot m_{s}\cdot \left({R_{s}}^{2}{{\;-\;r}_{s}}^{2}\right)=\frac{1}{2}\cdot 17.45\cdot (0.16^{2}\;-\;0.018^{2})=0.221\;\mathrm{kg}\cdot m^{2}$$
where R_s_ [m] is the radius of the flywheel shell, r_s_ [m] is the radius of the hole in the flywheel, m_s_ [kg] is the weight of the flywheel (m_s_ = 17.45 kg, read from the SolidWorks 3D modeler [39]).

The required braking torque M_B_ [N·m] can be expressed by Equation (9).
(9)$$M_{B}=J\cdot \varepsilon =\left(J_{M}+2\cdot J_{s}+J_{B}\right)\cdot \frac{2\cdot \pi \cdot n_{M}}{t_{b}}\;[N\cdot m]$$

When the electric motor [43] starts up to the speed n_M_ [min^−1^], which is set by the frequency converter [44] (nominal speed of the motor n_e_ = 710 min^−1^ = 12.3 s^−1^ [34]), it is disconnected from the power supply. For this case, Equation (10) can be determined, according to Figure 4c:(10)$$M_{B}{\;-\;M}_{u}{\;-\;M}_{r}=0\;\Rightarrow \;J\cdot \frac{2\cdot \pi \cdot n_{M}}{t_{b}}{\;-\;F}_{M}\cdot f\cdot \frac{l\cdot D}{a}{\;-\;M}_{r}=0\;[N\cdot m]$$
where M_r_ [N·m] is the torque expressing the system resistance (shaft, flywheel, and brake drum) against rotation (resistance in the bearings); the value was determined experimentally.

By changing the Equation (6) or (10), the value of friction coefficient f [-] (11) can be expressed.
(11)$$f=\frac{M_{M}\cdot a}{F_{M}\cdot l\cdot D}=\frac{J\cdot \frac{2\cdot \pi \cdot n_{M}}{t_{b}}{\;-\;M}_{r}}{F_{M}\cdot D\cdot \frac{l}{a}}=\frac{\left(J\cdot \frac{2\cdot \pi \cdot n_{M}}{t_{b}}{\;-\;M}_{r}\right)\cdot a}{F_{M}\cdot D\cdot l}{\;[\text{-}];\;f}_{1i}=\frac{M_{\mathrm{Mi}}\cdot a}{F_{Mi}\cdot l\cdot D}{\;[\text{-}];\;f}_{2i}=\frac{\left(J\cdot \frac{2\cdot \pi \cdot n_{M}}{t_{bi}}{\;-\;M}_{r}\right)\cdot a}{F_{\mathrm{Mi}}\cdot D\cdot l}\;[\text{-}]$$

Torque sensor 1 (type T4WA-S3 [35]) calibration was carried out as follows: The square of the sensor shaft 1 was clamped by the jaws of a bench vice 2 (see Figure 5a), and the square head of the sliding rod socket adapter socket wrench 3 [7] was inserted into the square hole in the sensor shaft 1. A steel tube 4 [8] (external diameter ϕ16 mm, wall thickness 1 mm, weight m_t_ = 0.37 kg·m^−1^) was inserted into the ϕ16 mm hole in the head of the sliding rod socket adapter socket wrench 2, to which a suspension nut with an M8 eye 5 (DIN 582) was attached using a screw. A weight 6 was hung on the suspension nut. The magnitude of the torque M_c_ [N·m], derived for sensor 1 calibration, can be calculated according to Equation (12).
(12)$$M_{c}=\left(m_{t}\cdot L_{t}+m_{z}\cdot L_{z}\right)\cdot g\;[N\cdot m]$$

The calibration of the force sensor 7 (type AST-250 kg [38]) (see Figure 5b) was carried out by hanging a weight 9 (known weight 232 N) on a hanger 8. As seen in Figure 6, the torque sensor cable and the force sensor cable, terminated with a D-Sub 9-pin plug, were connected to the module DS NET BR4 [45]. The DS GATE module [45] was connected to the PC (ASUS K72JR-TY131, ASUS, Suzhou, China), in which DEWESoft X2 SP5 software was installed, using the RJ45 connectors of the network cable [46].

## 3. Results

The measurements on the laboratory device (Figure 1 and Figure 2) were carried out using three procedures in a laboratory at the Department of Machine and Industrial Design, Faculty of Mechanical Engineering, VSB-Technical University of Ostrava.

The first measurements were performed to determine the resistance to rotation of the system M_r_ [N·m] of the laboratory device rotating parts; see Section 3.1 for details.

The second measurement detected the torque M_Mi_ [N·m] and the tensile force F_Mi_ [N] in the threaded rod connecting the pins in the upper parts of the double-jaw brake levers. From the known measured values M_Mi_ [N·m] and F_Mi_ [N], it was possible (using Equation (11)) to calculate the values of friction coefficient f_1i_ [-] of the brake lining against the steel casing of the brake disc rotating at revolutions n_M_ [min^−1^]. The specific procedure for determining the value of friction coefficient f_1i_ [-] is given in Section 3.2.

The third measurement was of the braking time t_bi_ [s] of the rotating parts system in the laboratory device, rotating at revolutions n_e_ [min^−1^]. System braking in the laboratory device was implemented by deriving the tensile force F_Mi_ [N] in the threaded rod connecting the pins of the two-jaw brake levers at the moment of electric current supply interruption into the electric motor power terminals. From the known dimensions of the two-jaw brake (see Figure 4), the moment of inertia of the laboratory device rotating parts (7), the moment M_r_ [N·m] (see Table 1), and the measured values n_M_ [min^−1^] and F_Mi_ [N], is possible to calculate the friction coefficient f_2i_ [-]; see Section 3.3 for more details.

### 3.1. Resistance Against the Rotation of the Laboratory Device Rotating Parts

The measurement of torque M_ri_ [N·m], expressing the resistance against the rotation of the system’s rotating parts (without the motor rotor), was carried out on a laboratory device (see Figure 3) with the electric motor 2 removed and brake jaws fully opened off the brake disc circumference 4. A mounting wrench was attached to the quadrilateral of the torque sensor 6, which was acted upon by a force that caused the system to rotate when a certain amount was exceeded. The generated moment, M_ri_ [N·m], as seen in Figure 7, acting on the torque sensor, was recorded by DEWESoft X2 SP5 measuring software [46].

The measurement of moment M_ri_ [N·m] was repeated 5 times under the same technical conditions; the values of M_ri_ [N·m] are shown in Table 1. The measurement results were statistically processed using Student’s distribution. For the risk α = 5% and the confidence coefficient P = 95%, the Student’s coefficient t_5%,5_ = 2.78 was chosen according to [47] for the number of repeated measurements n = 5. The arithmetic mean M_rs_ [N·m] of all measured values M_ri_ [N·m] and the extreme error κ_α,n_ = κ_%,5_[N·m] are shown in Table 1.

### 3.2. Friction Coefficients of the Brake Lining Against the Rotating Casing of the Brake Drum

When measuring starts, the drum brake jaws, fitted with BERAL 1126 [48] brake lining, are detached from the brake drum shell. Separation of the brake jaws from the brake drum is ensured by loosening a hexagonal nut screwed onto the threaded rod that connects the two brake levers. The threaded rod is broken, and both end parts of the fractured threaded rod are screwed with an AST-250 kg [38].

By applying electric current to the electric motor 1 AP 112M-S3 terminals (see Figure 8), a magnetic field is generated in the stator winding of the engine, which spins the rotor from zero revolutions to nominal revolutions n_eT_ [min^−1^] (13) (theoretically calculated revolutions, nominal revolutions n_e_ [min^−1^]).

The electric motor rotor rotates at revolutions neT [min^−1^], controlled by a frequency converter (YASKAWA VS-606 V7 [44]). A torque sensor 3 is installed between the electric motor shaft and the driven shaft 2 T4WA-S3 [35].
(13)$$n_{\mathrm{eT}}=n_{e}\cdot \;\frac{f_{c}}{f}{\;[\min}^{-1}]$$
where f [Hz] is the frequency of the alternating current in the distribution grid f = 50 Hz (Europe, f = 60 Hz USA), and fc [Hz] is the frequency set on a frequency changer.

The actual speed of the electric motor 1 n_M_ [min^−1^] was obtained from measuring speed sensor UNI-T UT373 4 [35]. A force sensor 5 (AST-250 kg) is installed in the upper part of both brake levers, which detects the force F_Mi_ [N]. Before starting the electric motor 1, the tensile force F_Mi_ [N] is derived by tightening the hexagonal nuts 6 in the threaded rod of the brake, interrupted, and fitted with a strain gauge force sensor 5.

At the moment when the rotor of the electric motor 1 rotates at the speed n_e_ [min^−1^], the force F_Mi_ [N] is sensed by the force sensor 5, and the torque M_Mi_ [N·m] by the torque sensor 4. According to (11), the value of friction coefficient f_1i_ [-] is calculated (see Table 2) for the brake lining—woven asbestos-free friction material BERAL 1126 containing brass fibers for increased resistance and high-temperature stability. The brake lining is made using resin additives and with the help of a three-axial structure [49].

The speed sensor UNI-T UT373 4 accurately detects the rotating motor rotor’s speed. When the speed of the motor rotor reaches the value of theoretical nominal speed n_eT_ [min^−1^], the hexagonal nut is tightened, which brings the ends of the brake levers closer to each other and thus increases the pressure of the brake lining against the shell of the brake drum.

The plugged electric motor 1 [34] rotates to the required theoretical revolutions n_eT_ [min^−1^] (13), set by using the YASKAWA VS-606 V7 frequency converter [44]. The actual speed of the electric motor n_M_ [min^−1^] is detected by the speed sensor 4 UNI-T. At this point, the brake shoes are not in contact with the brake disc.

Four repeated measurements under the same conditions, using the same DEWESoft DS-NET [45] measuring apparatus, obtained the values of force F_Mi_ [N] and torque M_Mi_ [N·m], listed in Table 2.

Table 2 shows the calculated value of friction coefficient f_1_ [-] in Equation (14) using Student’s distribution [47].
(14)$$f_{1}=f_{1s}\;\pm \;\chi_{\alpha ,n}=f_{1s}\;\pm \;\chi_{5\%,4}\;[\text{-}];\;\chi_{\alpha ,n}=t_{\alpha ,n}\cdot {s=t}_{5\%,4}\cdot s\;[\text{-}]$$
where f_1i_ [-] is the arithmetic mean of all (n [-]—number of repeated measurements) measured values f_1i_ [-], χ_α,n_ [-] is the extreme error, t_α,n_ [-] (t_5%,4_ = 3.18) is Student’s coefficient for risk α [%] (α = 5%) and confidence coefficient P [%] (P = 95%) [47], with [-]—sample standard deviation of the arithmetic mean.

Figure 9 presents the course of the tensile force F_Mi_ [N] measured by the force sensor [38] and the torque M_Mi_ [N·m] by the torque sensor [3] when the electric motor rotates at n_M_ [min^−1^] revolutions.

Table 3 shows the measured values of force F_Mi_ [N] and torque M_Mi_ [N·m] when the electric motor speed n_M_ [min^−1^] is controlled by the frequency converter with the set frequency values f_c_ = 30 Hz and f_c_ = 40 Hz.

Time recording of measured tensile force F_Mi_ [N] and torque M_Mi_ [N·m] when the electric motor rotates at speeds n_M_ [min^−1^] lower than the nominal speed of the electric motor [34] n_e_ = 710 min^−1^, controlled by a frequency converter [44], as shown in Figure 10.

Four repeated measurements under the same conditions, using the same DEWESoft DS-NET [46] measuring apparatus, obtained the values of force F_Mi_ [N] and torque M_Mi_ [N·m], listed in Table 4.

Time recording of measured values of tensile force F_Mi_ [N] and torque M_Mi_ [N·m] when the electric motor rotates at the speed n_M_ [min^−1^] identical to the nominal speed of the electric motor [34] n_e_ = 710 min^−1^, as shown in Figure 11.

### 3.3. Friction Coefficients of the Brake Lining Friction Against the Brake Disc Shell During the Disc’s Braking

The plugged electric motor [34] rotates to the required theoretical revolutions n_eT_ [min^−1^] (13), set by using the YASKAWA VS-606 V7 frequency converter [44]. The actual speed of the electric motor n_M_ [min^−1^] is detected by the speed sensor UNI-T UT373 [35]; see Table 5, Table 6 and Table 7. At this point, the brake shoes are not in contact with the brake disc.

The force sensor AST-250kg [38] is installed between the upper pins in the end parts of both levers of the double-jaw brake, detecting the force F_Mi_ [N]; see Figure 4. In the threaded rod of the brake, interrupted and fitted with a geometric force sensor [6] (see Figure 4), the force F_Mi_ [N] is derived by tightening the hexagonal nuts; (see Table 5, Table 6 and Table 7). The magnitude of force F_Mi_ [N] is detected by DEWESoft X2 SP5 [46].

After the required speed n_e_ [min^−1^] of the electric motor and a required tensile force F_Mi_ [N] in the threaded rod (DEWESoft X2 SP5 records the time course of force F_Mi_ [N] values) are achieved, the electric motor is disconnected from the power supply (measurement time t_1i_ [s]; see Figure 12 and Figure 13). Due to the friction force N_1_ = N_2_ [N] (3) and (5), a uniform reduction in the system revolutions takes place until the system is entirely stopped (measuring time t_2i_ [s]; see Figure 12 and Figure 13). The braking time t_bi_ [s] is calculated from the measured times t_1i_ [s] and t_2i_ [s].

From the known brake dimensions a [m], l [m], and D [m] (see Figure 4), a total moment of system inertia J [kg·m^2^] (7), the moment of system resistance against rotation M_ri_ [N·m] (see Section 3.1), and the braking time t_bi_ [s], the value of friction coefficient f_2i_ [-] can be calculated according to (11), when moving.

As shown in Table 5, DEWESoft X2 SP5 software obtained start times t_1i_ [s] and end times t_2i_ [s] of rotating parts system braking (revolutions n_M_ = 148.5 min^−1^ for f_c_ = 10 Hz and n_M_ = 298.4 min^−1^ for f_c_ = 20 Hz) in the laboratory device and the magnitude of force F_Mi_ [N] when the braking of laboratory device rotating parts starts.

**Table 5 sensors-24-08078-t005:** The magnitude of F_Mi_ [N] at the start of braking t_1i_ [s] of the rotating system at speeds n_M_ = 148.5 min^−1^ and 298.4 min^−1^, the time of the end of braking t_2i_ [s].

n_M_ [min^−1^]	F_E_ [N]	t_1i_ [s]	t_2i_ [s]	t_bi_ [s]	f_2i_ [-]	n_M_ [min^−1^]	F_E_ [N]	t_1i_ [s]	t_2i_ [s]	t_bi_ [s]	f_2i_ [-]
148.5	37.0 ^1^	18.53 ^2^	19.57 ^3^	1.04	0.369	298.4	99.4 ^4^	17.39 ^5^	18.44 ^6^	1.05	0.351
	33.9	16.17	17.31	1.14	0.355		95.5	18.87	19.98	1.11	0.344
	29.7	14.88	15.94	1.06	0.326		92.3	18.42	19.41	0.99	0.390
	35.1	17.40	18.38	0.98	0.310		93.3	18.56	19.45	0.89	0.325
			Σ f_2i_ [-]		1.360				Σ f_2i_ [-]		1.410
			f_2i_ = Σ f_2i_/i [-]		**0.340**				f_2i_ = Σ f_2i_/i [-]		**0.353**
			χ_5%,4_ [-]		**0.054**				χ_5%,4_ [-]		**0.043**

^1^ see Figure 9a, ^2^ see Figure 12a, ^3^ see Figure 12b, ^4^ see Figure 9c, ^5^ see Figure 12c, ^6^ see Figure 12d.

In Table 6, DEWESoft X2 SP5 software-obtained start times t_1i_ [s] and end times t_2i_ [s] of rotating parts system braking in the laboratory device (revolutions n_M_ = 448.5 min^−1^ for f_c_ = 30 Hz and n_M_ = 598.4 min^−1^ for f_c_ = 40 Hz) are presented, along with the magnitude of force F_Mi_ [N] when braking of the laboratory device rotating parts starts.

**Table 6 sensors-24-08078-t006:** The magnitude of F_Mi_ [N] at the start of braking t_1i_ [s] of the rotating system at speeds n_M_ = 448.5 min^−1^ and 598.4 min^−1^, the time of the end of braking t_2i_ [s].

n_M_ [min^−1^]	F_E_ [N]	t_1i_ [s]	t_2i_ [s]	t_bi_ [s]	f_2i_ [-]	n_M_ [min^−1^]	F_E_ [N]	t_1i_ [s]	t_2i_ [s]	t_bi_ [s]	f_2i_ [-]
448.5	69.1 ^1^	24.18 ^2^	26.13 ^3^	1.95	0.378	598.4	81.1 ^4^	24.06 ^5^	26.43 ^6^	2.37	0.366
	65.9	23.07	25.15	2.08	0.368		87.5	22.19	24.63	2.44	0.338
	51.9	19.04	21.45	2.41	0.380		84.7	23.19	25.84	2.65	0.321
	71.6	26.31	28.13	1.82	0.392		88.3	25.84	26.32	2.26	0.359
			Σ f_2i_ [-]		1.518				Σ f_2i_ [-]		1.384
			f_2i_ = Σ f_2i_/i [-]		**0.380**				f_2i_ = Σ f_2i_/i [-]		**0.346**
			χ_5%,4_ [-]		**0.015**				χ_5%,4_ [-]		**0.038**

^1^ see Figure 10a, ^2^ see Figure 13a, ^3^ see Figure 13b, ^4^ see Figure 10c, ^5^ see Figure 13c, ^6^ see Figure 13d.

Figure 13 presents the course of measured tensile force F_Mi_ [N] and the torque M_Mi_ [N·m] values when the electric motor rotates at a speed of n_M_ = 448.5 min^−1^ (Figure 13a,b) or 598.4 min^−1^ (Figure 13c,d). The start time t_1i_ [s] and end time t_2i_ [s] of the braking system of laboratory device rotating parts, at the frequency f_c_ = 30 Hz on the frequency converter, are given in Figure 13a,b. The start time t_1i_ [s] and end time t_2i_ [s] of the braking system of the laboratory device rotating parts, at the frequency f_c_ = 40 Hz on the frequency converter, are given in Figure 13c,d.

As shown in Table 7, the DEWESoft X2 SP5 software obtained the start times t_1i_ [s] and end times t_2i_ [s] of the rotating parts system braking (revolutions n_M_ = 748.2 min^−1^ for f_c_ = 50 Hz) of the laboratory device. The magnitude of force F_Mi_ [N] when braking of the laboratory device’s rotating parts starts is presented.

**Table 7 sensors-24-08078-t007:** The magnitude of F_Mi_ [N] at the start of braking t_1i_ [s] of the rotating system at speed n_M_ = 748.2 min^−1^, the time of the end of braking t_2i_ [s].

n_M_ [min^−1^]	F_E_ [N]	t_1i_ [s]	t_2i_ [s]	t_bi_ [s]	f_2i_ [-]
748.2	88.9 ^1^	20.82 ^2^	23.63 ^3^	2.81	0.359
	72.3 ^4^	29.63 ^5^	32.61 ^6^	2.98	0.341
	83.2	33.79	36.84	3.05	0.350
	85.3	29.92	33.13	3.21	0.329
			Σ f_2i_ [-]		1.379
			f_2i_ = Σ f_2i_/i [-]		**0.345**
			χ_5%,4_ [-]		**0.022**

^1^ see Figure 11a, ^2^ see Figure 14a, ^3^ see Figure 14b, ^4^ see Figure 11b, ^5^ see Figure 14c, ^6^ see Figure 14d.

Figure 14 presents the course of the measured tensile force F_Mi_ [N] and the torque M_Mi_ [N·m] values when the electric motor rotates at speed n_M_ = 748.5 min^−1^. Figure 14a,b give the start time t_1i_ [s] and end time t_2i_ [s] of the braking system of the laboratory device rotating parts at the frequency f_c_ = 50 Hz on the frequency converter.

## 4. Discussion

Crane brakes are used in technical practice for two purposes: either to stop sliding and rotational movement (parking brakes) or to maintain the load’s movement at the required speed (if the device is not self-locking or started in another way, starting brakes).

With double-jaw brakes, the braking effect comes from friction generated by the pressure of the brake lining attached to the brake shoes against the brake drum shell.

The brake friction coefficient is also one of the key parameters of a vehicle’s braking performance. The coefficient fundamentally affects the optimization of braking systems and the vehicle braking performance.

Bartolomeo et al. [50] present an experimental and numerical analysis of friction-induced vibrations arising from frictional contact between two bodies in relative motion. The sliding contact was reproduced within a mechanical system characterized by simple dynamics to better distinguish between the system’s dynamic response and the broadband induction coming from the contact. In this article, the authors research the effects of some parameters, especially relative speed, roughness, and normal load, on the size and frequency content of induced vibrations and compare the experimental measurements and simulation results.

Because the magnitude of the friction coefficient cannot be measured directly, a laboratory device was designed and created in a laboratory at the Department of Machine and Industrial Design, Faculty of Mechanical Engineering, VSB-Technical University of Ostrava, on which the torque and tensile force were detected. The DEWESoft DS-NET measuring apparatus subtracted the necessary data from the measurement records in the DEWESoft X2 SP5 software environment, and the friction coefficient was calculated.

The values obtained by measuring the friction coefficient of the brake lining against the casing of a rotating (revolutions n_M_ [min^−1^]) brake drum (see Section 3.2) were the highest at the lowest revolutions n_M_ < n_e_ [min^−1^] and the lowest at the highest revolutions n_M_ = n_e_ = 710 min^−1^.

At the set frequency f_c_ = 10 Hz on the frequency converter, with the parts of the laboratory device rotating at speeds n_M_ = 148.5 min^−1^ (see Table 2), the friction coefficient f1(10 Hz) = 0.469 ± 0.031 value was calculated.

At f_c_ = 20 Hz (n_M_ = 298.4 min^−1^; see Table 2), the value of friction coefficient f_1(20Hz)_ = 0.357 ± 0.010 was calculated, which corresponds to 0.76% of the value f_1(10Hz)_ [-].

At f_c_ = 30 Hz (n_M_ = 448.5 min^−1^; see Table 3), the value of friction coefficient f_1(30Hz)_ = 0.365 ± 0.077 was calculated, which corresponds to 0.78% of the value f_1(10Hz)_ [-].

At f_c_ = 40 Hz (n_M_ = 598.4 min^−1^; see Table 3), the value of friction coefficient f_1(40Hz)_ = 0.357 ± 0.032 was calculated, which corresponds to 0.76% of the value f_1(10Hz)_ [-].

At f_c_ = 50 Hz (n_M_ = 598.4 min^−1^; see Table 4), the value of friction coefficient f_1(40Hz)_ = 0.333 ± 0.081 was calculated, which corresponds to 0.71% of the value f_1(10Hz)_ [-].

The measurements on the laboratory device were carried out when the brake disc casing surface was clean and dry. In practice, no supply of oil or lubricants to the surface of the brake drum case is allowed due to a drastic reduction in friction coefficient.

Braking time measurement on the laboratory device was carried out at different speeds n_M_ [min^−1^] of its rotating parts: the electric motor rotor, flywheel, brake disc, and shaft. After the rotating parts had started up to the required angular speed, the electric motor power supply was disconnected, and the brake was put into operation by the tensile force F_Mi_ [N] in the threaded rod, sensed by the sensor. The measuring software recorded the total braking time tbi [s].

Table 5 shows the calculated values of friction coefficient f_2(10Hz)_ = 0.340 ± 0.054 and f_2(20Hz)_ = 0.353 ± 0.043 for the actual revolutions n_M(10Hz)_ = 148.5 min^−1^ and n_M(20Hz)_ = 148.5 min^−1^ of the laboratory device rotating parts. At frequency f_c_ = 20 Hz on the frequency converter, the average value of the friction coefficient f_2(20Hz)_ is 104% of the value f_2(10Hz)_.

Table 6 shows the calculated values of friction coefficient f_2(30Hz)_ = 0.380 ± 0.015 and f_2(40Hz)_ = 0.346 ± 0.038 for the actual revolutions n_M(30Hz)_ = 448.5 min^−1^ and n_M(40Hz)_ = 598.4 min^−1^ of the laboratory device rotating parts. At frequency f_c_ = 30 Hz on the frequency converter, the average friction coefficient f_2_(30Hz) value is 112% of f_2(10Hz)_. At frequency f_c_ = 40 Hz on the frequency converter, the mean value of friction coefficient f_2(40Hz)_ is 102% of the value f_2(10Hz)_.

Table 7 shows the calculated values of friction coefficient f_2(50Hz)_ = 0.345 ± 0.022 for the actual revolutions n_M(50Hz)_ = 748.2 min^−1^ of the laboratory device rotating parts. At f_c_ = 50 Hz on the frequency converter, the mean value of friction coefficient f_2(50Hz)_ is 102% of the value f_2(10Hz)_.

The difference in the average friction coefficient values between the brake lining and the rotating brake drum casing f1(iHz) in Section 3.2 and the values (f_2_) in Section 3.3 can be attributed to the challenges in obtaining completely accurate braking time readings tbi [s] when recording the braking of the laboratory device’s rotating parts using the DEWESoft program. Substituting slightly different braking times into Equation (12) results in a change in the calculated friction coefficient value in the brake lining against the casing of the rotating brake drum.

The arithmetic average of the calculated friction coefficients value in the brake lining against the casing of the rotating brake drum f_1(iHz)_ [-], listed in Table 2, Table 3 and Table 4, is f_1_ = 0.376. The arithmetic average of the calculated friction coefficients value in the brake lining against the casing of the rotating brake drum f_2(iHz)_ [-], listed in Table 5, Table 6 and Table 7, is f_2_ = 0.353. The value f_2_ = 0.353 is 93.9% of f_1_ = 0.376.

It is also undesirable for water to reach the surface of the brake drum, as mainly during the occasional use of a double-jaw brake, the dry friction could change into semi-dry friction or, in extreme cases, even liquid friction could occur. The area between the dry and wet state of the contact brake surfaces is not determined from a physical point of view. It is also unclear how thick a layer of water or damp dirt is needed to fundamentally change the friction coefficient value in the contact surfaces of a double-jaw brake.

Double-jaw brakes are friction brakes. By the friction of the brake lining against the brake drum surface, minute particles are separated, and heat is generated by changing the kinetic energy of all sliding and rotating masses into friction and heat. The heat generated during braking, especially in high-performance machinery, dries the moisture/water, but in occasional braking mode or with a selected short braking time, declaring the value of dry friction coefficient is impossible when brake contact surfaces are wet.

Experimental investigation and determination of the boundary between the dry and wet states of the brakes’ contact surfaces stimulate further research. In practice, it is also desirable to determine a minimum thickness of the liquid layer from the measurement results, at which point the dry friction coefficient would still be considered for technical calculations.

The construction design of the described laboratory device and its execution in the laboratory were carried out to obtain the values of the friction coefficient of brake linings for drum brakes with external shoes for their subsequent verification and comparison with the values declared by the manufacturer. The manufacturer often states the friction coefficient of brake linings only as a value corresponding to the operating temperature. Still, this coefficient is unknown for the contact conditions (presence of pollution, humidity, presence of lubricant, and oil) under which the brakes are commonly operated in actual conditions.

In the bachelor’s form of study at the Faculty of Mechanical Engineering, VSB Technical University of Ostrava, the device will be used to obtain values of the static friction coefficient for different brake linings and different states of contact surfaces, which will then be assessed and compared.

## 5. Conclusions

The implemented measurements aimed to design a laboratory device on which the most accurate value of friction coefficient could be experimentally obtained for the Beral 1126 type brake lining glued to the brake shoes of a double-jaw brake under dry operating conditions. When writing this article, the authors did not consider comparing the values of friction coefficients in different friction materials used as brake linings from the beginning.

Brakes, regardless of their design, serve to reduce the speed or stop a moving load (service brakes) and to secure a stationary load against unwanted movement (parking brake). The braking effect is achieved by friction between fixed and rotating parts. The kinetic energy turns into a thermal one, which needs to be dissipated into the atmosphere.

The calculated average value of the friction coefficient in brake lining type Beral 1126 when the brake disc is rotating, calculated from the parameters measured by the force and torque sensors on the laboratory device (see Section 3.2), is f_1_ = 0.376.

The measured friction coefficients of the brake lining type Beral 1126 result in the average value of friction coefficient f_2_ being 0.353, based on the braking time measured on the laboratory device according to the procedure described in Section 3.3.

Double-acting jaw brakes are widespread components of drive units of continuously and intermittently operating transport devices. Due to the smaller acting torques, the brake drum is designed as a flexible connector between the drive motor and the gearbox. The braking effect of shoe brakes comes from clamping the jaws (usually loosely mounted on pins passing through holes in the brake levers) on the outer shell of the brake drum with a compression spring or weight. A single-acting electromagnet or an electrohydraulic brake release performed brake release.

From the data obtained by measuring sensors (see Table 1, Table 2, Table 3, Table 4, Table 5, Table 6 and Table 7), it can be stated that the magnitude of the friction coefficient of the brake lining can be determined, with sufficient accuracy, on the designed laboratory device. Future research aims to implement measurements that allow for the determination of friction coefficient values in various friction materials used as brake linings under different operating conditions—when wet or when contaminated with oil or dust.

## Figures and Tables

**Figure 1 sensors-24-08078-f001:**
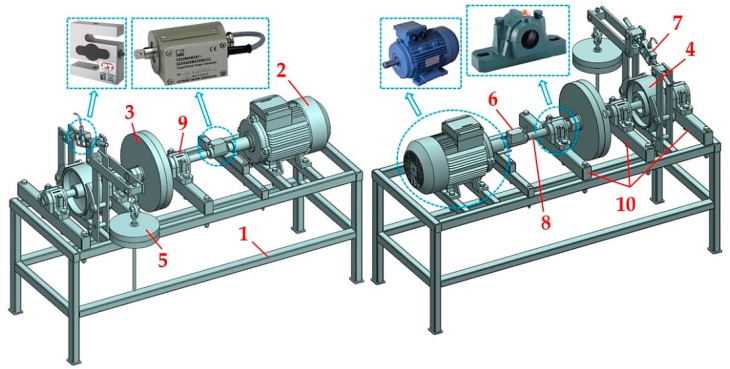
Laboratory device detecting the friction coefficient of the brake lining. 1—steel structure; 2—electric motor; 3—flywheel; 4—double-jaw brake; 5—weights; 6—torque sensor [35]; 7—force sensor [38]; 8—shaft; 9—plummer block [36]; 10—Jäckl 60 × 40.

**Figure 2 sensors-24-08078-f002:**
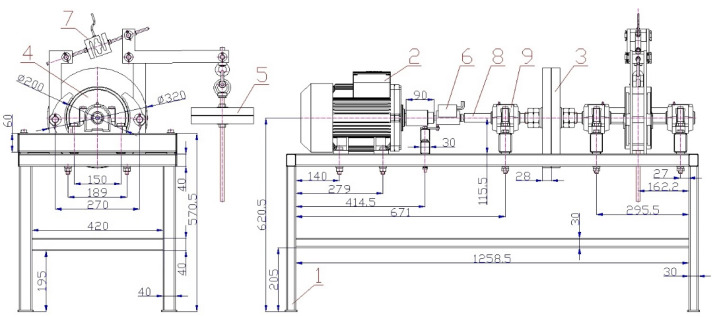
Two-dimensional laboratory device sketch. 1—steel structure; 2—electric motor; 3—flywheel; 4—double-jaw brake; 5—weights; 6—torque sensor [35]; 7—force sensor [38]; 8—shaft; 9—plummer block [36].

**Figure 3 sensors-24-08078-f003:**
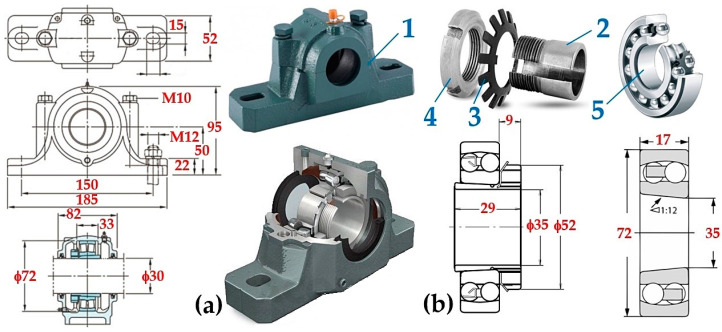
(**a**) Plummer block SN 507 [36], (**b**) clamping sleeve A207X [41]. 1—plummer block; 2—clamping sleeve; 3—KM nut KM7 [42]; 4—MB washer MB7; 5—tilting ball bearing 1207 K.

**Figure 4 sensors-24-08078-f004:**
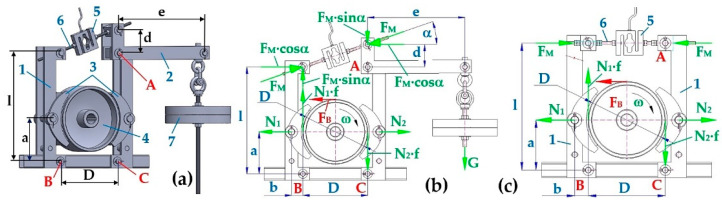
Two-jaw disc brake. (**a**) Basic dimensions of the brake, (**b**) forces acting in the brake pins, (**c**) tensile force acting in the threaded rod. 1—brake arm; 2—brake lever; 3—brake shoes; 4—brake drum; 5—force sensor [38]; 6—threaded rod; 7—weight.

**Figure 5 sensors-24-08078-f005:**
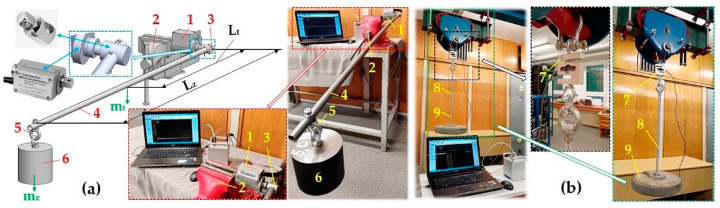
Calibration of (**a**) T4WA-S3 torque sensor, (**b**) AST-250 kg force sensor. 1—sensor T4WA-S3; 2—bench vice; 3—head of assembly sliding rod socket adapter socket wrench; 4—steel pipe; 5—suspension nut; 6—weight; 7—sensor AST-250 kg; 8—hinge; 9—weight.

**Figure 6 sensors-24-08078-f006:**
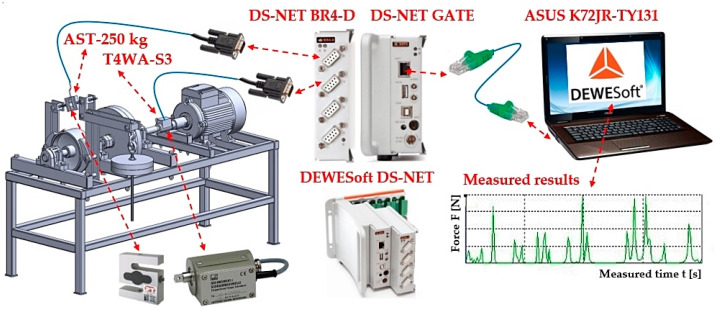
A measuring chain is a sequence of interconnected devices and equipment that enables the detection and processing of measured signals.

**Figure 7 sensors-24-08078-f007:**
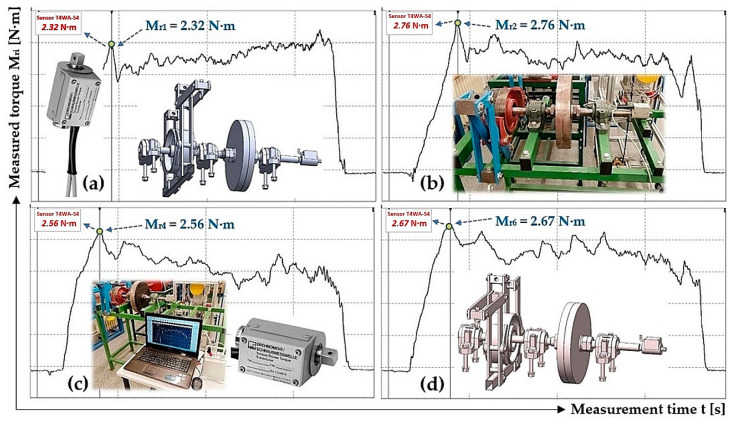
The time course of torque M_ri_ [N·m], expressing the system resistance against rotation, was measured on a laboratory device. (**a**) M_r1_ = 2.32 N·m, (**b**) M_r2_ = 2.76 N·m, (**c**) M_r4_ = 2.56 N·m, (**d**) M_r6_ = 2.67 N·m.

**Figure 8 sensors-24-08078-f008:**
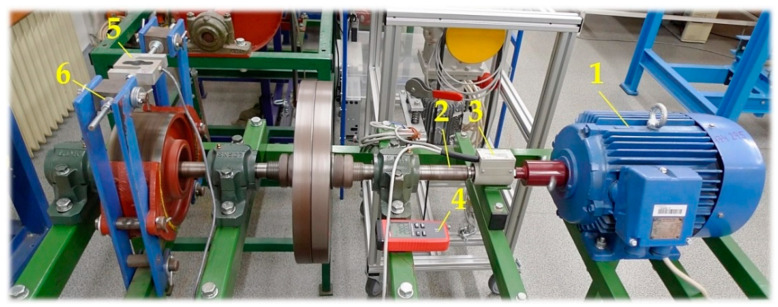
A laboratory device was created to determine the friction coefficient f_1i_ [-] of the brake lining in motion and the friction coefficient f_2i_ [-] of the brake lining when the rotating parts of the laboratory device are braked. 1—electric motor; 2—driven shaft; 3—torque sensor; 4—speed sensor; 5—force sensor; 6—hexagonal nut.

**Figure 9 sensors-24-08078-f009:**
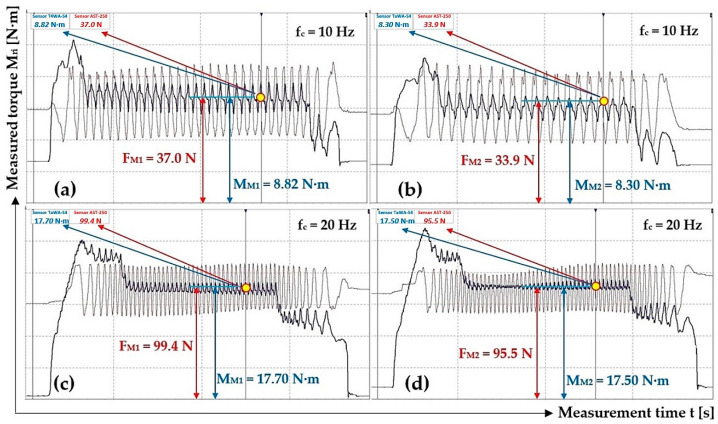
Time recording of tensile force F_Mi_ [N] and torque M_Mi_ [N·m] measured on a laboratory device for f_c_ [Hz] (**a**,**b**) 10, (**c**,**d**) 20.

**Figure 10 sensors-24-08078-f010:**
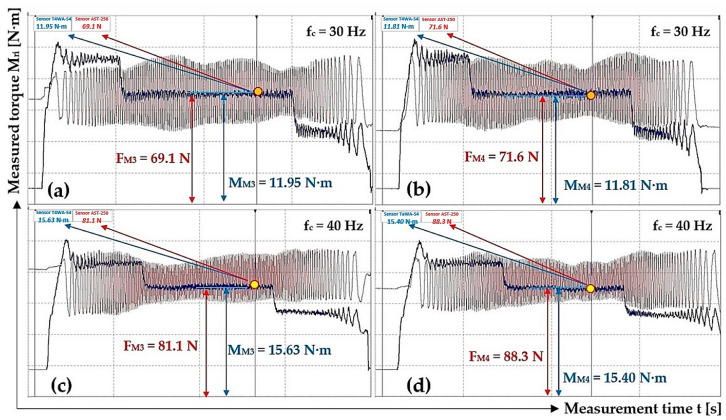
Tensile force F_Mi_ [N] and torque M_Mi_ [N·m] course, measured on a laboratory device for f_c_ [Hz] (**a**,**b**) 30, (**c**,**d**) 40.

**Figure 11 sensors-24-08078-f011:**
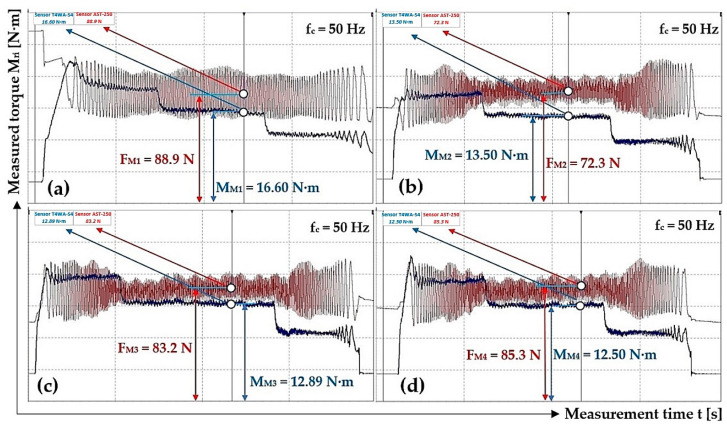
Tensile force F_Mi_ [N] and torque M_Mi_ [N·m] course, measured on laboratory device for f_c_ = 50 Hz. (**a**) F_M1_ = 88.9 N, M_M1_ = 16.60 N·m; (**b**) F_M2_ = 72.3 N, M_M2_ = 13.50 N·m; (**c**) F_M3_ = 83.2 N, M_M3_ = 12.89 N·m; (**d**) F_M4_ = 85.3 N, M_M4_ = 12.50 N·m.

**Figure 12 sensors-24-08078-f012:**
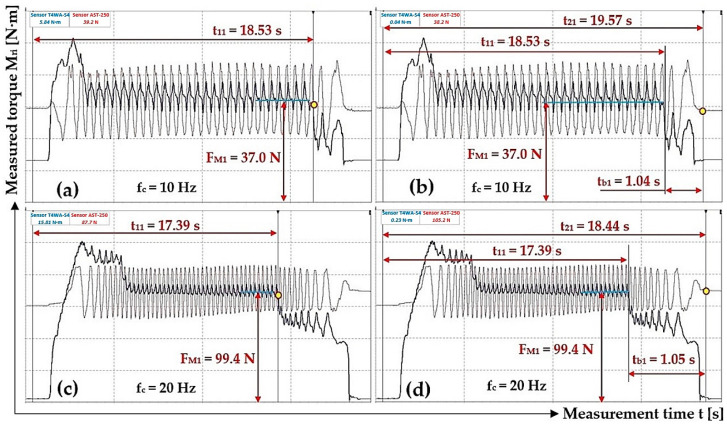
Tensile force F_Mi_ [N] and torque M_Mi_ [N·m] course measured on the laboratory device during the braking of the rotating parts system. (**a**) f_c_ = 10 Hz, t_11_ = 18.53 s; (**b**) f_c_ = 10 Hz, t_21_ = 19.57 s; (**c**) f_c_ = 20 Hz, t_11_ = 17.39 s; (**d**) f_c_ = 20 Hz, t_21_ = 18.44 s.

**Figure 13 sensors-24-08078-f013:**
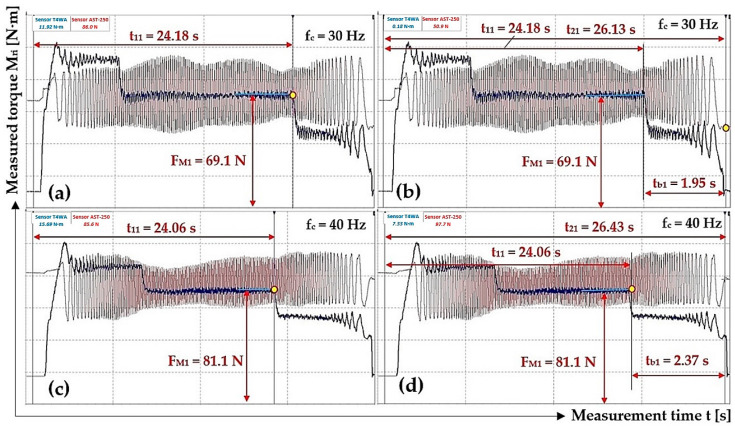
Tensile force F_Mi_ [N] and torque M_Mi_ [N·m] course measured on the laboratory device during braking of the rotating parts system. (**a**) f_c_ = 30 Hz, t_11_ = 24.18 s; (**b**) f_c_ = 30 Hz, t_21_ = 26.13 s; (**c**) f_c_ = 40 Hz, t_11_ = 24.06 s; (**d**) f_c_ = 40 Hz, t_21_ = 26.43 s.

**Figure 14 sensors-24-08078-f014:**
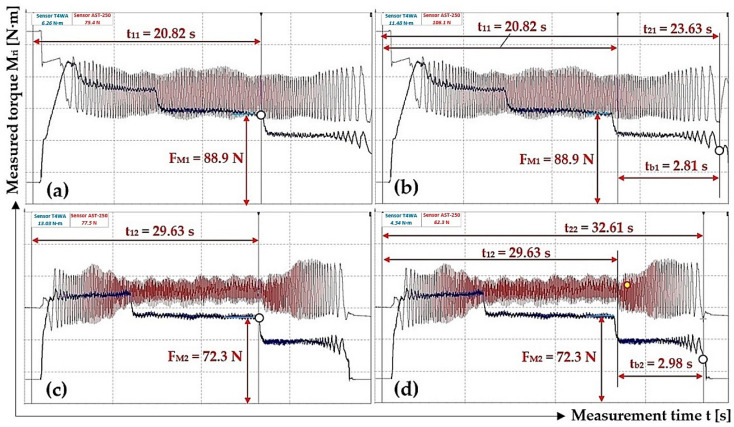
Tensile force F_Mi_ [N] and torque M_Mi_ [N·m] course, measured on the laboratory device during braking of the rotating parts system of the laboratory device at f_c_ = 50 Hz. (**a**) t_11_ = 20.82 s, (**b**) t_21_ = 23.63 s, (**c**) t_12_ = 29.63 s, (**d**) t_22_ = 32.61 s.

**Table 1 sensors-24-08078-t001:** Measurement of the torque expressing the system’s resistance to rotation M_r_ [N·m] on laboratory device.

M_ri_ [N·m]	2.32 ^1^	2.76 ^2^	2.56 ^3^	2.81	2.67 ^4^	Σ M_ri_ [N·m]	13.12	M_rs_ [N·m]	**2.62**	κ_%,5_ [N·m]	**0.26**

^1^ see Figure 7a, ^2^ see Figure 7b, ^3^ see Figure 7c, ^4^ see Figure 7d.

**Table 2 sensors-24-08078-t002:** Measured values of the force F_Mi_ [N] and the torque M_Mi_ [N·m] on the laboratory device when the rotor rotates at speeds n_e_ = 148.5 min^−1^ and 298.4 min^−1^.

f_c_ [Hz]	n_M_ [min^−1^]	F_Mi_ [N]	M_Mi_ [N·m]	f_1i_ [-]	f_c_ [Hz]	n_M_ [min^−1^]	F_Mi_ [N]	M_Mi_ [N·m]	f_1i_ [-]
10	148.5	37.0 ^1^	8.82 ^1^	0.470	20	298.4	99.4 ^3^	17.70 ^3^	0.351
		33.9 ^2^	8.30 ^2^	0.482			95.5 ^4^	17.50 ^4^	0.361
		29.7	7.27	0.482			92.3	16.60	0.354
		35.1	7.88	0.442			93.3	17.13	0.362
			Σ f_1i_ [-]	1.876				Σ f_1i_ [-]	1.428
			f_1s_ = Σ f_1i_/i [-]	**0.469**				f_1s_ = Σ f_1i_/i [-]	**0.357**
			χ_5%,4_ [-]	**0.031**				χ_5%,4_ [-]	**0.010**

^1^ see Figure 9a, ^2^ see Figure 9b, ^3^ see Figure 9c, ^4^ see Figure 9d.

**Table 3 sensors-24-08078-t003:** Measured values of force F_Mi_ [N] and torque M_Mi_ [N·m] on the laboratory device when the motor rotor rotates at speeds n_M_ = 448.5 min^−1^ and 598.4 min^−1^.

f_c_ [Hz]	n_M_ [min^−1^]	F_E_ [N]	M_Mi_ [N·m]	f_1i_ [-]	f_c_ [Hz]	n_M_ [min^−1^]	F_E_ [N]	M_Mi_ [N·m]	f_1i_ [-]
30	448.5	51.9	11.37	0.432	40	598.4	84.7	15.56	0.362
		65.9	12.11	0.362			87.5	15.23	0.343
		69.1 ^1^	11.95 ^1^	0.341			81.1 ^3^	15.63 ^3^	0.380
		71.6 ^2^	11.81 ^2^	0.325			88.3 ^4^	15.40 ^4^	0.344
			Σ f_1i_ [-]	1.460				Σ f_1i_ [-]	1.429
			f_1s_ = Σ f_1i_/i [-]	**0.365**				f_1s_ = Σ f_1i_/i [-]	**0.357**
			χ_5%,4_ [-]	**0.077**				χ_5%,4_ [-]	**0.032**

^1^ see Figure 10a, ^2^ see Figure 10b, ^3^ see Figure 10c, ^4^ see Figure 10d.

**Table 4 sensors-24-08078-t004:** Measured values of the force F_Mi_ [N] and the torque M_Mi_ [N·m] on the laboratory device when the motor rotor rotates at speed n_M_ = 748.2 min^−1^.

f_c_ [Hz]	n_M_ [min^−1^]	F_E_ [N]	M_Mi_ [N·m]	f_1i_ [-]
50	748.2	88.9 ^1^	16.60 ^1^	0.368
		72.3 ^2^	13.50 ^2^	0.368
		83.2 ^3^	12.89 ^3^	0.305
		85.3 ^4^	12.50 ^4^	0.289
			Σ f_1i_ [-]	1.330
			f_1s_ = Σ f_1i_/I [-]	**0.333**
			χ_5%,4_ [-]	**0.081**

^1^ see Figure 11a, ^2^ see Figure 11b, ^3^ see Figure 11c, ^4^ see Figure 11d.

## Data Availability

The original contributions presented in the study are included in this article; further inquiries can be directed to the corresponding author.

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
