# Peer review of "Signal Detection by Sensors and Determination of Friction Coefficient During Brake Lining Movement"

_sensors, 2024, doi:10.3390/s24248078_

Round 1
Reviewer 1 Report
Comments and Suggestions for Authors
Tips for Authors:
The article discusses important issues from the point of view of the automotive industry. It complements existing research in this area. The test stand itself seems to be very helpful when making such an assessment.
To make the work more useful for other researchers, it is worth supplementing it with the following content:
Number of samples tested, time and duration of the test.
The method of detecting signals and their recording must be discussed in more detail. It is worth pointing out the mathematical dependencies in this regard.
What is the error range? It is also worth mentioning the accuracy and precision of the measurements.
The research was based on the specific material Beral 1126 - what was the purpose?
In the summary, it is worth indicating the direction and prospects for further research in relation to other materials and their alloys.
After making changes, it is worth publishing the work.
Reviewer 2 Report
Comments and Suggestions for Authors
The paper with title "Signals Detection by Sensors and Determination of Friction Coefficient During Brake Lining Movement" is processed at a high level. - The abstract is related to the content and objectives of the scientific article
- Introduciton contains a suitable research of current studies in the issue
- I consider the Materials and Methods section to be the supporting part, where the authors described in detail the designed measuring device with pressure and torque sensors. The part is supplemented with a calculation model.
- The Result section appropriately describes the processing with an example of measurements at the proposed measuring station.
- The conclusion is processed scientifically correctly.
- In the referencees do not use SK and CZ language, but EN translation / ref 41, 42, 43/
Reviewer 3 Report
Comments and Suggestions for Authors
In this manuscript, the author obtained a relatively accurate friction coefficient of the Beral 1126 type brake lining when bonded with a double-jaw brake in a dry environment by using a laboratory device that can monitor and record the signal change process. Under the same conditions, mainly through four repeated experiments with the same DEWESoft DS-NET, the force and torque were obtained and then the average value of the friction coefficient of the Beral 1126 type brake lining was calculated. By accelerating the rotating parts to a certain angular velocity and then disconnecting the power supply to start the brake, the total braking time recorded by the measurement software was detected. Based on the measured braking time, the average friction coefficient of the Beral 1126 type brake lining was measured, thus proving that the experimental equipment has relatively sufficient accuracy in measuring the friction coefficient of the brake lining under the condition that the surface of the brake disc housing is clean and non-humid. The specific comments are as follows:
1. The writing logic of the abstract is not concise enough. The abstract of the article does not cover the key information of the text, fails to effectively extract the core points, and lacks a close connection with the main body of the article. It is necessary to accurately summarize the essence of the article.
2. In the introduction part, the differences and innovation points of this research compared with previous studies can be further elaborated to highlight the unique value of this research.
3. Please improve the quality of the pictures to make them clearer.
4. Please add the names of the parts indicated by the labels in Figure 8 to be consistent with the legend above.
5. On page 12, line 333, an extra ")" is added after "[44]", please delete it.
6. In the content on lines 416 - 423 on page 15, the symbol ")" is missing. Please supplement it.
7. In the conclusion part, it can be more specific about how future research will be carried out in view of the current limitations of the research.
8. The brake friction coefficient is one of the key parameters for vehicle braking performance. It is recommended that the author include content in the introduction and conclusion about the friction coefficient concerning the optimization of braking systems and vehicle braking performance. On one hand, this highlights the necessity and significance of the study. On the other hand, it emphasizes the supportive significance of this research for subsequent work. Related works can refer to the paper “Parametrical experimental and numerical analysis on friction-induced vibrations by a simple frictional system”, “Dynamic modeling and brake stick-slip vibration analysis of a vehicle-track system with disc-pad nonlinear frictions”.
Comments on the Quality of English LanguageEnglish expression can be further improved.
Reviewer 4 Report
Comments and Suggestions for Authors
This paper presents a laboratory device to measure the friction coefficient of the brake lining against the casing of a rotating brake drum. The content of the work is sufficient, but there are still some issues need to be addressed before its possible publication.
Major Comments:
1. The abstract and the title should be revised to clearly show the main topic and novelty of this work. The introduction section should clearly illustrate how this work is novel from the existing works in the field.
2. In section 2, why is two modelling software used to create the sketch in Fig. 2?
3. The axes and labels are missing for Figures 7, 9, 10, 11, 12, 13 and 14.
4. The results are not logically presented. The current demonstration is a sequential description of figures and tables. The meaning and necessity of some tests should be given for a better understanding of the work.
Minor comment:
1. The numbering of the equations is not aligned.
2. In line 215, the right parenthesis is missing for “(type AST-250 kg [38]”. Line 267, “[44])”.
3. Table 1 is disorganized, that possibly causing confusion to readers.
4. Line 416 to line 423 can be listed in a table if necessary.
Comments on the Quality of English LanguageMinor editing of English language required.
Round 2
Reviewer 4 Report
Comments and Suggestions for Authors
The authors have addressed all my concerns.